# Simulating the Benefits of Nature Exposure on Cognitive Performance in Virtual Reality: A Window into Possibilities for Education and Cognitive Health

**DOI:** 10.3390/brainsci12060725

**Published:** 2022-05-31

**Authors:** Michel T. Léger, Said Mekari

**Affiliations:** 1Faculté des Sciences de L’éducation, Université de Moncton, Moncton, NB E1A 3E9, Canada; michel.leger@umoncton.ca; 2Department of Family Medicine, Université de Sherbrooke, Sherbrooke, QC J1K 2R1, Canada; 3Centre de Formation Médicale du Nouveau-Brunswick, Pavillon J.-Raymond Frenette, Moncton, NB E1A 7R1, Canada

**Keywords:** virtual reality, nature walks, education, memory, cognitive performance

## Abstract

Purpose: This one-group pretest–posttest, designed within a subject study, looks to compare the effects of an outdoor nature walk (ONW) to those of a virtual nature walk (VRW) on memory and cognitive function. Implications are discussed for education as well as for the world of virtual reality. Methods: Sixty-four healthy university students were asked to complete an ONW and a VRW, which was created using 3D video of the same nature trail used for the ONW. The VRW condition involved a five-minute walk on a treadmill, while wearing a virtual reality mask (Oculus, San Francisco, USA) that projected a previously recorded three-dimensional capture of the same nature walk they experienced outdoors. Both experimental conditions lasted approximately 5 min and were counterbalanced between participants. A Digit Span Test (Digit) for working memory and a Trail Test (TMT) for executive function were administered to all study participants, immediately before and after each type of walk. Results: For executive function testing (Trail Making Test), our results demonstrate that both the ONW and VRW condition improved the TMT time, when compared to a baseline (ONW 37.06 ± 1.31 s vs. 31.75 ± 1.07 s, *p* < 0.01 and VRW 36.19 ± 1.18 s vs. 30.69 ± 1.11 s, *p* < 0.01). There was no significant difference between the ONW and VRW groups. Similarly, for the Digit memory task, both conditions improved compared to the baseline (ONW 54.30 ± 3.01 vs. 68.4 ± 2.66, *p* < 0.01 and VRW 58.1 ± 3.10 vs. 67.4 ± 2.72, *p* < 0.01). There was a difference at the baseline between the ONW and VRW conditions (54.3 ± 3.01 vs. 58.1 ± 3.10, *p* < 0.01), but this baseline difference in memory performance was no longer significant post exercise, between groups at follow-up (68.4 ± 2.66 vs. 67.4 ± 2.72, *p* < 0.08). Conclusions: Our results suggest that both a virtual reality protocol and a nature walk can have positive outcomes on memory and executive function in younger adults.

## 1. Introduction

Despite documented positive effects associated with nature interactions [1,2,3,4], there are fewer and fewer opportunities to experience nature worldwide. It seems that the way in which we experience the world today is, increasingly, through our exposure to screens, namely from using cell phones, videogames, and computers. The trend towards a more technologically rich world is present in all aspects of modern twentieth-century life, from health care, to entertainment, industry, education, and physical activity [5]. Advances in physical (e.g., artificial intelligence, driverless cars, and 3D printing), digital (e.g., the Internet of things) and biological (e.g., synthetic biology and bio-printing) technologies are changing the way we learn and live, driving economic competitiveness and social development [6]. In fact, given the fourth industrial revolution [7], a new form of higher education is emerging that looks at teaching, research, and service in a different way, one where university (and public schooling, for that matter) is interdisciplinary and comprised of virtual-learning spaces [6]. Twenty-first-century learners, even, seem to prefer educational environments that are highly interactive and include augmented or virtual reality (VR), as part of the lesson plan [8]. Educational research is still unclear, as to whether teaching in such virtual environments is beneficial to learning, though research does point to an increased interest in virtual reality, as an educational tool among students [9,10,11].

In recent years, virtual reality exercise has been recognised as a new approach to promote physical activity and health behaviours [12] and is becoming, increasingly, used in health promotion as well as in other fields not related to health, such as education. VR is operationally defined as a “computer-generated display that allows or compels the user (or users) to have a sense of being present in an environment other than the one they are actually in and to interact with that environment” ([13] p. 25). The present article deals with non-immersive VR, where young adults experienced a simulated nature walk in virtual reality and were asked to perform pre and post cognitive tests, for memory and executive function. Virtual reality environments can range from non-immersive to fully immersive, according to the degree with which a user is isolated from their physical surroundings, while interacting with the virtual environment [14]. In our study, participants interacted with nature virtually, from a visual and auditory perspective only; they were not asked to interact with the virtual world through manipulation, for example.

Although there are, still, unanswered questions on the effectiveness of teaching in virtual environments, research seems clearer, when it comes to the impacts of nature experiences on human cognitive function and mental health [15,16]. Outdoor educational spaces, also, seem to have benefits for learning and general wellbeing [17,18,19]. For instance, cognitive research has shown that school-aged children can demonstrate improved performance on mental tasks, such as memory, when such tasks are undertaken in a natural setting [17]. Other studies have shown that children, who attend pre-schools with natural outdoor play areas, scored better when tested for executive functioning [18]. As for the practice of physical activity in the presence of nature, known as green exercise [20], it has been postulated that it can have greater value than exercising in a gym, for instance, when it comes to preventing disease and enhancing general health [21]. A review of the literature, on the cognitive effects of nature exposure, revealed that most studies focus less on cognitive effects and, mainly, report on the health benefits of nature exposure, such as decreased hypertension and allergic response [19]. From a psychological perspective, more research, examining the potential executive function benefits of nature exposure for learning, seems warranted.

Given the popularity of technologies today, as well as the reported benefits of learning in a natural environment, this pilot study explores the possibilities associated with blending both of these approaches to learning. More specifically, the present contribution responds to the lack of research on the potential cognitive effects of nature walks versus a nature walk in virtual reality for adult learners, namely in university students. We hypothesise that participating university students would perform as well on memory and executive tests when they experience a virtual reality nature walk, compared to an actual forest nature walk.

## 2. Materials and Methods

### 2.1. Participants

A total of 81 undergraduate students (46 females, ages 20 to 24; 35 males, ages 21 to 25) gave their written consent, to participate in this study. Of the total number of participants, 81, three female participants withdrew midway through, citing time constraints, resulting in a total sample of 78. All participants reported being healthy and having normal-to-corrected vision, following a pre-study interview. During these preliminary interviews, none of the participants reported having a history of neurological disorder, colour blindness, involuntary tremors, epilepsy, or drug/alcohol-dependency problems. The protocol was reviewed and approved by the Institutional Research Ethics Board at the Université de Moncton (ethics certificate # 1718-105) and was conducted in accordance with the Declaration of Helsinki.

### 2.2. Study Design

This study is designed as a one-group pretest–posttest intervention, with cognitive-performance data collected before and after the applied intervention. Before taking part in the data-collection session, participants were asked to complete the consent form, and researchers asked them questions about their general health, specifically in regards to potential issues related to eyesight, balance, and general mobility. The study consisted of two types of data-collection situations (experimental conditions), each lasting a total of approximately 45 min. The experimental conditions consisted of either an Outdoor Nature Walk (ONW) or a Virtual Reality Walk (VRW). All data collection took place within a ten-week span, in the summer of 2021, and cognitive tests were administered prior to (pre) and after (post) both walking conditions. The selection of the cognitive tests is based on previous studies, which reveal a positive relationship between an acute bout of aerobic exercise, executive function, and an academic achievement test [22]. Since the same participants were asked to undergo both experimental conditions, counterbalancing measures were implemented, to allow for more control over potential confounding variables.

In terms of procedures and protocol, for the ONW, participating students were brought to a nearby forest and directed to walk for five minutes, along a straight nature trail. Immediately before and after the walk, they performed two cognitive tests, specifically the Digit Span Memory Test and the Trail Making Test (Part B only). A table and chair were set up at the trailhead, allowing the participants to perform these tests while comfortably seated. One month after the outdoor nature walk, the same participants returned to an indoor laboratory for the VRW condition. There, they performed the same two cognitive tests, this time before and after a virtual reality walk, involving a five-minute walk on a treadmill, while wearing a virtual reality mask (Oculus, San Francisco, CA, USA) that projected a previously recorded three-dimensional capture, of the same nature-walk they experienced outdoors. The treadmill speed was set at five kilometers per hour, to match the moderate walking pace suggested during the outdoor nature walk. Furthermore, time in virtual reality was limited to approximately five minutes, in part to limit the potential onset of “simulator sickness”, an occasional side effect of virtual reality [23]. Noise-cancelling headphones with nature sounds were, also, worn, to drown out ambient noises (i.e., the mechanical sounds of the treadmill) and simulate the typical sounds heard during the ONW. In the case of VRW, the simulated virtual reality walk was created using 3D video of the same nature trail used for the ONW. The trail was filmed professionally, using a *Humaneyes Technologies* XR 3D VR 180°/2D 360° 5.7 K camera, by Vuze (Israel), operated by a trained videographer. The treadmill speed was set by the researchers at four kilometers per hour, which was considered to be a reasonably common leisurely walking pace. It is, also, worth mentioning that, as measured by a heart-rate monitor (Polar Electro, Kempele, Finland), there was no significant difference in exercise intensity between the ONW and the VRW.

### 2.3. Cognitive Assessment

#### 2.3.1. Trail Making Test (TMT), Part B

Before and after each experimental condition, all participants completed only Part B of the Trail Making test. Participants were given the following Trail Making Test instructions: Please take the pencil and draw a line from one number to one letter, in ascending order. Start at 1 [point to the number], then go to A [point], then go to 2 [point], then go to B, and so on. Please try not to lift the pencil as you move from one number to the next. Work as quickly and accurately as you can. Participants were encouraged to correct their errors, and this was included in the total time to complete. The speed at which all the numbers were connected was measured in seconds. This test was chosen to measure cognitive flexibility or switching ability. Part B differs from Part A, specifically in that it assesses more complex factors of motor control and perception [24]. We chose to administer Part B only, since this test is better suited to measure executive function, according to Arbuthnott and Frank (2000), which is, specifically, what we wanted to investigate in this study, along with memory, as expressions of cognitive function. In both experimental conditions, different versions of Trail Test B were administered, to avoid performance bias, where participants could score higher on the posttest, due to repetition.

#### 2.3.2. Digit Span Test (Digit)

The Digit Span test was administered using standard testing procedures, as described by Wechsler [25] and applied in numerous research articles involving this cognitive evaluation method [26,27,28]. In the Digit Forward part, participants listened to a series of digits (numbers) read to them at a rate of 1 digit per second, by a research assistant. Following presentation of the digit series, participants were asked to report the digits, in the order presented. The digits ranged from the numbers 1–9, and the length of the series ranged from 2–9 digits, with two trials at each series length. When participants missed both trials at a given series length, testing was discontinued. Participants then completed the Digit Backwards test. The procedure for this test was identical to the Digit Forward, except to report the digits in reverse order. Memory-span scores for Digits Forward and Backward were recorded, as the number of items in the longest series correctly recalled. An overall score was compiled, by adding the forward and backward scores to determine a “standard score” and using a comparative table to, ultimately, produce a percentile equivalent [29]. As was the practice in Part B of the Trail Making Test, we provided a different version of the Digit Span Test in the pre and post components of our design, thus attempting to avoid performance bias, resulting in a higher score on the post element due to repetition.

### 2.4. Statistical Analysis

Standard Statistical methods were used for the calculation of means and standard deviations. The normal Gaussian distribution of the data was verified by the Shapiro–Wilk test, and the homoscedasticity was verified by a modified Levene’s test. The compound symmetry, or sphericity, was checked by Mauchly’s test. When the assumption of sphericity was not met, the degree of freedom of the F-ratios was adjusted, according to the Greenhouse–Geisser procedure, when the epsilon correction factor was <0.75, or according to the Huynh-Feldt procedure, when the epsilon correction factor was >0.75. For both the cognitive tasks (TMT and Digit), a 2 Groups (ONW and VRW) × 2 Time (Pre vs. Post walk) mixed ANOVA was conducted. All post-hoc t-tests were Bonferroni-corrected, for multiple comparisons. For the purposes of this study, the statistical significance level (alpha level) was set at 0.05, which is in keeping with most studies undertaken in the field of education, and the probability level (*p* value) was calculated using SPSS (version 26, 2019). In fact, the significance level was set at *p* < 0.05 for all analyses.

## 3. Results

### 3.1. Trail Making Test

The statistical analysis of the Trail Making Test (TMT) revealed a main effect of time F(1,78) = 31.4, *p* < 0.01, where participants were quicker to complete the TMT post-exercise (36.9 ± 1.1 s pre vs. 31.2 ± 1.0 s post). Analyses do not show a main effect of condition. Statistical results can be found in Table 1.

### 3.2. Digit Span Test

The statistical analysis of the Digit Span test (also found in Table 1) revealed a main effect of time F(1,78) = 78.1, *p* < 0.01, where participants had higher scores post-exercise (66.1 ± 2.5 vs. 56.33 ± 2.9). Analyses, also, shows a main effect of condition F(1,78) = 9.01, *p* = 0.04, where participants in the VRW condition expressed higher scores when compared to the ONW condition (62.8 ± 2.8 vs. 59.7 ± 2.68). Paired t-tests confirmed a Bonferroni-corrected significant difference, at rest, between the VRW and the ONW conditions (58.2 ± 3.1 vs. 54.3 ± 3.1, *p* < 0.02). This statistical difference seems to no longer exist, after both walking conditions (67.4 ± 2.7 vs. 64.9 ± 2.6, *p* = 0.08).

## 4. Discussion

The purpose of this exploratory study was to determine the possible cognitive effects of a nature walk in virtual reality, versus an actual nature walk, for a sample of university students. Based on the existing literature, we hypothesised that participating university students would perform as well on memory and executive tests, when they experience a virtual reality nature walk, compared to an actual forest nature walk. Our findings suggest that both a simulated nature walk in virtual reality and an actual outdoor nature walk can both have positive outcomes on memory and executive function, in younger adults. Specifically, performance on the cognitive tasks was significantly improved post-walk, in both conditions.

Regarding cognitive performances in natural settings, studies have shown that individuals can demonstrate improved performance on mental tasks, such as memory, when such tasks are undertaken in a natural setting [17]. Other studies have shown that children who attend pre-schools with natural outdoor play areas scored better when tested for cognitive functioning [18]. Similar to our findings, Rogerson et al. (2016) found that for both the outdoor and indoor conditions, there was a statistically significant condition x time interaction effect [30]. Miera et al. (2014), also, fail to report significant differences in attention scores, between the outdoor and indoor exercise conditions [30]. Further studies, by Rider et al. (2019), failed to find an influence of walking on memory in either the nature, the urban, or the indoor environments [31]. After a comprehensive review of the green literature, Lahart et al. (2019) did not find evidence that exercising in outdoor or virtual green environments offers superior benefits to exercising indoors, without exposure to nature [20]. Therefore, if the exercise benefits do not differ from one condition to the other, this could explain very well why we saw cognitive improvements in both conditions.

Studies on exercise and cognition have demonstrated evidence that both short-term memory (ES = 0.26) and long-term memory (ES = 0.52) can be enhanced, after an acute bout of physical activity [32]. Reviews by Loprinzi et al. (2019) and Blomstrand et al. (2020), also, show that the effect of acute exercise on short-term memory and working memory may be more pronounced for younger adults. Other studies, also, discuss the importance of exercise intensity, when enhancing memory [33,34]. Similar to our methodology, some meta-analytic studies suggest that shorter exercise duration (<20 min) may be optimal in enhancing short-term memory [35]. In addition to improvements in memory post exercise, our study has, also, demonstrated an increase in executive function. Although the results of individual studies are mixed, Chang et al. (2012) published a meta-analysis, which concluded that a single bout of exercise (aerobic or resistance) can produce a small but reliable cognitive benefit, in executive function [36]. For their part, Samani et al. (2018) showed that a 10 min bout of aerobic exercise benefits executive function, as measured in younger adults. These authors attribute their findings to an exercise-based increase in arousal within the frontoparietal networks supporting executive function [37]. Other studies have shown a relationship between exercise, executive function, and an academic achievement test. Hillman et al. (2009) revealed that a single acute bout of aerobic exercise could improve executive function and neural responses. This improvement, also, led to an increase in academic achievement test. They conclude that aerobic exercise might serve as a cost-effective means for improving specific aspects of academic achievement and enhancing cognitive function [22]. As such, documented benefits of nature exposure, for memory and higher cognitive functioning, can be made available for all students, as a planned pedagogical strategy for test preparation or general wellbeing. Furthermore, specific to environmental education, such findings support the design of inclusive pedagogical activities, to teach about nature by immersing oneself in nature, whether it be by using virtual reality, augmented reality, simulation, or, potentially, even games. We do not advocate replacing real outdoor experiences, but, rather developing alternate experiential platforms for students who cannot get outdoors, for example. Moreover, if the virtual nature experiences in question are designed around pedagogical games, for instance, why not use them as teaching tools for all students, from time to time?

Despite its limited scope and exploratory nature, our study contributes to the lack of research in this area and, in turn, to the understanding of virtual reality’s potential benefits for adults in terms of learning, as indicated by improved cognitive functioning. Though our study involves a relatively small number of participants, and though we fully recognise further research is necessary, in order to speak in more general terms, our findings do point to interesting implications for education, regarding cognition, particularly executive function and memory. Given the exploratory nature of this pilot study, no control group data were collected. Instead, we structured the study in such a way, as to measure and compare levels of cognitive performance before and after two variations of a nature walk, within a single group. We fully understand that choosing not to add a control group to the methodological design represents an important limitation and recommend that future studies consider a design structure that involves a control group. Another limitation of our study is that cognitive tasks were performed relatively close to each other, thus making it impossible to rule out a possible learning effect. Numerous studies that have repeatably measured cognition (memory and executive function), throughout the day or during a short period of time, report no learning effect/performance bias [38,39,40]. That said, our data indicate that an increased frequency of the test did not yield significantly better accuracy, demonstrating that there was not a substantial learning effect associated with the way this task was administered.

## 5. Conclusions

Data collected in this study seem to indicate that university students (in the case of this particular study) can perform just as well on a memory test (Digit Span Test) after experiencing a walk in a virtual reality environment than they can after experiencing an actual outdoor forest walk. When tested for more complex cognitive function (Trail B Test), the same university students, also, seemed to perform as well in both the natural forest and virtual reality settings. Such results point to the relevance and importance of further research in virtual reality simulations and has implications for education as well as for a variety of other fields, such as geriatric mental health care or the world of virtual reality, as a gaming environment. Despite the preliminary nature of our pilot study, the results do add to the growing discussion on the potential benefits of nature exposure in virtual reality, focusing specifically on cognitive effects (i.e., enhanced memory, processing abilities), an aspect rarely addressed in the literature.

In education, if the results from this study can be replicated in a larger experimental study, the conclusions expressed here could have significant implications, especially for students who are not able to venture into the natural environment because of physical challenges, for example. Giving such students a, comparably, beneficial nature walk experience, even though it is in a virtual environment, represents a concrete example of applied inclusive education, as it enables all learners, no matter their circumstances, to benefit from strategies that can potentially lead to enhanced memory and cognitive performance. Finally, from a non-formal education perspective, our results could, also, have applications for the elderly, who might benefit, cognitively, from a virtual nature walk, when unable to venture into the forest due to physical limitations. If virtual nature walks can have effects on cognition, similar to those documented during outdoor walks, elderly people unable to get outside could, also, benefit from nature’s influence on cognitive performance.

## Figures and Tables

**Table 1 brainsci-12-00725-t001:** Cognitive response to both exercise protocols.

Experiential Environments	Pre				Post			
	Digit Test (pctl)		Trail B (s)		Digit Test (pctl)		Trail B (s)	
	Mean	SE	Mean	SE	Mean	SE	Mean	SE
Nature Walk	54.3	3.1	37.1	13.7	64.9	2.6	31.7	8.1
Virtual Reality Walk	58.2	3.1 ^a^	36.1	10.9	67.4	2.7	30.69	6.8

s = seconds/pctl = percentile score/SE = standard error ^a^ different from “VRW pre”.

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
