# Peer review of "Simulating the Benefits of Nature Exposure on Cognitive Performance in Virtual Reality: A Window into Possibilities for Education and Cognitive Health"

_brainsci, 2022, doi:10.3390/brainsci12060725_

Round 1

Reviewer 1 Report

Thank you for the opportunity to review your manuscript. Whilst the concept is interesting, I found the study design inappropriate in answering your question, as well as the need to tone down your results. Limitations do not address the fact that this effect could be a practice effect, and I would suggest using alternative memory/cognitive tasks to further corroborate your results. 

Author Response

Thank you for your comments and suggestions. All modifications have been incorporated in the new document.

Reviewer 2 Report

The paper addresses an interesting topic, and the overall structure is easy to follow. However, I have a few major concerns and several minor concerns regarding the paper.

Major concerns

1. Most importantly, the study does not include a control group. The authors contribute the improved cognitive performance to the virtual and outdoor nature walks. The lack of a control group compromises such conclusions as the authors cannot distinguish the effects of nature exposure from other factors that could lead to improved performance, such as the learning effect/performance bias or just taking a break from everyday routines. The authors are clearly aware of the potential for a learning effect/performance bias and have included procedures to reduce the effect, but have not referenced any literature supporting that these procedures are sufficient to exclude the learning effect/performance bias. Such literature needs to be included in order to support the conclusions of the paper. If this literature does not exist, the study should not be published.

2. The literature review in the introduction is insufficient. There are several papers on the cognitive benefits of nature and green exercise that should be included. Although a bit outdated, the following reviews might be a good place to start to identify original papers relevant to your paper:

Bratman, G. N., Hamilton, J. P., & Daily, G. C. (2012). The impacts of nature experience on human cognitive function and mental health. Year in Ecology and Conservation Biology, 1249, 118–136. https://doi.org/10.1111/j.1749-6632.2011.06400.x

Lahart, I., Darcy, P., Gidlow, C., & Calogiuri, G. (2019). The Effects of Green Exercise on Physical and Mental Wellbeing: A Systematic Review. Int J Environ Res Public Health, 16(8). https://doi.org/10.3390/ijerph16081352

3. The discussion is mostly a presentation of research on related topics. I suggest rewriting the discussion with an emphasis on explaining the findings of the study and its implications. The reviews mentioned above might be a good place to start to identify theories that might explain your findings.

Minor concerns

4. I suggest you replace the term “3D filming” with “360 video” throughout the paper. I believe that is the common term used to describe video recordings used in VR.

5. This is your definition of VR: “VR is operationally defined as digital technology wherein sensory experiences, (e.g., visual, auditory, touch, and scent stimuli) are artificially created, prompting users to manipulate the objects within virtual environment”. As I understand, there are no way for participants to manipulate objects within the virtual environment that you created, and thus, you did not use VR by your own definition. Is suggest replacing this definition of VR with something that is more appropriate for your study.

6. Try to avoid repeat information in Materials and Methods (e.g. that cognitive performance data are collected pre and post is mentioned several times, and the pre-study interview as well).

7. Was the speed of the treadmill matched with the speed of the video? This is a potential strength or limitation as discussed here:

Litleskare, S., MacIntyre, T. E., & Calogiuri, G. (2020). Enable, Reconnect and Augment: A New ERA of Virtual Nature Research and Application. Int J Environ Res Public Health, 17(5). https://doi.org/10.3390/ijerph17051738

8. The mention of the “null hypothesis” in the conclusion section should be changed to “hypothesis”.

Author Response

Major concerns

  1. Most importantly, the study does not include a control group. The authors contribute the improved cognitive performance to the virtual and outdoor nature walks. The lack of a control group compromises such conclusions as the authors cannot distinguish the effects of nature exposure from other factors that could lead to improved performance, such as the learning effect/performance bias or just taking a break from everyday routines. The authors are clearly aware of the potential for a learning effect/performance bias and have included procedures to reduce the effect, but have not referenced any literature supporting that these procedures are sufficient to exclude the learning effect/performance bias. Such literature needs to be included in order to support the conclusions of the paper. If this literature does not exist, the study should not be published.

Answer 1: Answer 1: Reviewers bring up a very important topic that we feel was not fully discussed in our paper. Belinger et al. (2005) examined practice effects in 28 healthy adults.  Neuropsychological tests were administered in a fixed order for 6 weeks. Results reveal that significant improvements were observed for all tests except the Trail B test. Our methodology is also similar to a paper published by Rogerson et al. (2016), who measured cognition (Digit test) after 15 minutes of cycling indoor or outdoor. An earlier paper from the same team (Rogerson et al., 2015) also measured cognition before and after 15 min of aerobic exercise on a treadmill using virtual reality. Another literature that supports the exclusion of learning effect/performance bias is the acute effects of interrupting prolonged periods of sitting with physical activity on cognition field. Chueh et al., (2022) have recently published a systematic review on this topic. These studies measure cognitive function numerous times throughout the day and do not show a learning effect or performance bias. Bergouignan et al. (2016) found that in 30 adults, there was no improvement of the Trail Making Test when regularly tested over a 6-hour day. Wennberg et al. (2016) have also measured cognition throughout a 7-hour day in overweight and obese individuals. They do not report any improvement on working memory task (the digit is a working memory task) in this population. Also, our data indicated that an increased frequency of the test did not yield significantly better accuracy, demonstrating that there was not a substantial learning effect associated with the way this task was administered.

Nevertheless, we agree with the reviewer that this is a limitation of our study. Therefore, we have included the following paragraph in our limitation section of the discussion:

Another limitation of our study is that cognitive tasks were performed relatively close to each other and so we can’t rule out a learning effect. Ourdata indicated that an increased frequency of the test did not yield significantly better accuracy, demonstrating that there was not a substantial learning effect associated with the way this task was administered.

Rogerson, M.; Gladwell, V.F.; Gallagher, D.J.; Barton, J.L. Influences of green outdoors versus indoors environmental settings on psychological and social outcomes of controlled exercise. Int. J. Environ. Res. Public Health 2016, 13, 363.

Rogerson M, Barton J. Effects of the Visual Exercise Environments on Cognitive Directed Attention, Energy Expenditure and Perceived Exertion. Int J Environ Res Public Health. 2015 Jun 30;12(7):7321-36. doi: 10.3390/ijerph120707321. PMID: 26133125; PMCID: PMC4515658.

Beglinger LJ, Gaydos B, Tangphao-Daniels O, Duff K, Kareken DA, Crawford J, Fastenau PS, Siemers ER. Practice effects and the use of alternate forms in serial neuropsychological testing. Arch Clin Neuropsychol. 2005 Jun;20(4):517-29. doi: 10.1016/j.acn.2004.12.003. PMID: 15896564.

Wennberg P, Boraxbekk CJ, Wheeler M, Howard B, Dempsey PC, Lambert G, Eikelis N, Larsen R, Sethi P, Occleston J, Hernestål-Boman J, Ellis KA, Owen N, Dunstan DW. Acute effects of breaking up prolonged sitting on fatigue and cognition: a pilot study. BMJ Open. 2016 Feb 26;6(2):e009630. doi: 10.1136/bmjopen-2015-009630. PMID: 26920441; PMCID: PMC4769400.

Bergouignan A, Legget KT, De Jong N, Kealey E, Nikolovski J, Groppel JL, Jordan C, O'Day R, Hill JO, Bessesen DH. Effect of frequent interruptions of prolonged sitting on self-perceived levels of energy, mood, food cravings and cognitive function. Int J Behav Nutr Phys Act. 2016 Nov 3;13(1):113. doi: 10.1186/s12966-016-0437-z. PMID: 27809874; PMCID: PMC5094084.

Chueh TY, Chen YC, Hung TM. Acute effect of breaking up prolonged sitting on cognition: a systematic review. BMJ Open. 2022 Mar 15;12(3):e050458. doi: 10.1136/bmjopen-2021-050458. PMID: 35292487; PMCID: PMC8928248.

  1. The literature review in the introduction is insufficient. There are several papers on the cognitive benefits of nature and green exercise that should be included. Although a bit outdated, the following reviews might be a good place to start to identify original papers relevant to your paper:

Answer 2: We thank the reviewer for the comments, and we have changed our introduction to include a much more updated literature review.

  1. The discussion is mostly a presentation of research on related topics. I suggest rewriting the discussion with an emphasis on explaining the findings of the study and its implications. The reviews mentioned above might be a good place to start to identify theories that might explain your findings.

Answer 3: Once again we thank the reviewer for the comments. The discussion has been re-written completely. We hope this will satisfy the reviewer’s advice.

Minor concerns

All minor concerns have been modified in the text.

Reviewer 3 Report

Review

Overall, I understand why the topic is interesting and how a study with university students could contribute to this research area. I also really think that the approach to use VR in school and learning contexts, and with students who might be unable to experience something in natural environments, is very interesting and I’d like to read more about that. With regard to the specific study that was performed here, I’m not really sure if the findings support the claims made in the discussion section. Until the discussion section, I was not even really sure what the main aim of the study was – this should be much clearer from the very beginning and that’s why I would suggest to rework the introduction to better map the study (or the part of the study that the authors would like to publish here). From what I have read, I’m not sure that the authors can claim that ‘our study contributes [..] to the understanding of virtual reality’s potential benefits for adults in terms of learning, as indicated by improved cognitive functioning’. The results for working memory and task switching did not differ significantly after the two types of walks, meaning that virtual reality walks do not improve cognitive functioning more than real walks, right? Moreover, without a control group, we cannot know for sure if there is no cognitive adaptation that makes participants generally better after doing a test again after a short period of time – so it might not even be the walk (regardless of VR or not) that made the difference but rather the fact that participants knew the task better, could use a specific strategy or whatsoever. Additionally, no real learning task was used and the cognitive measures are only subsets of tests used to measure cognitive skills. This should definitely be discussed, also in terms of limitations.

Here are my comments for each subsection (section always mentioned before the comments that I have for that section):

  1. Introduction

Maybe the first sentence could be rephrased to ‘namely from using cell phones and computers, or playing videogames’ – somehow using videogames sounds wrong to me

twenty-first century learners

‘in a variety of the field of health promotion’ – does not make sense in the context

There is a misplaced comma before a bracket on page 2 (‘sensory experiences, (e.g.,…))

Does ‘with most studies focusing less on cognitive effects and mainly reporting on’ require the on at the end? I would have skipped it

‘this paper looks to respond to’ does not seem to me like the most elegant way to phrase here – maybe better ‘aims to/seeks to fill this gap’

Natural environment is often used in the singular, I would chance it to the plural ‘benefits of learning in natural environments’

I think the last paragraph of the intro should better introduce the study and give more details on why the study was conducted, what it measured and what the hypotheses were. Why should a walk generally improve working memory or executive functioning and how is this related to virtual reality? It also doesn’t seem warranted to say ‘memory and executive tests’ as digit span and the trail making test do not fully cover memory and executive function but just a small portion, namely phonological working memory and cognitive flexibility. This should be made clear. I’m afraid there is also no real info on these two skills although they are at the core of the study – by neglecting the specific skills, it seems as though the main aim of the study was just to show that virtual reality does the same as natural environments and the two tasks were randomly choses without prior hypotheses.  

  1. Materials and Methods

2.2. Study design

Not sure if you really say ‘data collected at pre and at post intervention’ I would have skipped the ‘at’

‘potential issues related TO eyesight, balance and general mobility’

I’m not sure what it means that potential confounding variables like temporal position were included? Is it just that you randomized who did which walk first?

‘participating students were brought to a forest nearBY

‘noise cancelling headphoneS with nature sounds were worn to drowned out ambient noise’

2.3. Cognitive Assessment

‘all participants completed Part B of the Trail Making Test only’

Not sure the verb is the best choice here: ‘which is specifically what we looked to explore in this study’ – maybe investigate? explore? research?

I know that researchers often tend to make a common working memory score, which I do understand if a lot of tests are administered. However, there is evidence that while digit span forward and backward overlap, they seem to measure different components of the working memory system – digit span forward is considered to measure capacity of the phonological loop (easy working memory, so to say) and digit span backward is considered to measure capacity of the central executive (complex working memory where material is not only retained, but also manipulated). Since you only did the Trail Making Test and the two digit span measures, I would rather not mix the two together, unless you show that their correlation is so high that it makes sense to lump them into one variable.

Is there any reason why digit span scores were used in percentile ranks and trail making test scores not?

2.1  2.4 Statistical Analysis

I think there is some confusion in the ANOVA description – I guess Group and Time were not both within-subject factors?

3.1.1. Trail Making Test

‘Analysis do not show’ – either the analysis or analyses; same problem in later sections

  1. Discussion

The first paragraph describes briefly the results of the study but doesn’t provide any interpretation as to why a nature walk in and outside of VR should positively impact cognition? Is it by reducing stress and increasing attention (clearing your mind)? How would that impact both phonological working memory and task switching?

I’m a little confused why some references are provided in brackets, others as numbers.

Maybe this should have been explained in the introduction, but I’m not exactly sure how the present findings relate to the few studies that are mentioned here – was really memory improved in the Mancuso et al. study? Long-term memory? Working memory? Visuo-spatial? Were those also pre-post design studies?

‘playing outdoors can boosted cognitive functioning’

‘though’ is used twice within two sentences

‘per say’ probably means ‘per se’?

I’m not sure as to whether we can really interpret the findings of the study as support that learning outcomes or processes are improved through VR. I don’t really agree that learning was measured.

Author Response

  1. Introduction

Thank you for your comments. The introduction has been re-written to better facilitate the understanding of our topic.

  1. Materials and Methods

2.2. Study design

Thank you for your comments. All modifications were made to the manuscript.

2.3. Cognitive Assessment

Is there any reason why digit span scores were used in percentile ranks and trail making test scores not?

Answer: Reviewer bring up a very good point. We decided to use these measures since unpublished work from our lab has done the same and we wanted to be consistent with the results presentation. Other studies have also presented these results in this way.

2.1  2.4 Statistical Analysis

Changes were made based on the reviewer’s comments.

3.1.1. Trail Making Test

The modifications have been made to these sections.

  1. Discussion

The discussion has been completely re-written to incorporate comments from the reviewer.

Round 2

Reviewer 2 Report

Nicely done. Most of my concerns has been addressed and I think the quality of the paper has improved. I have one final suggestion. In your response to my previous comments you explained in detail why you are not overly concerned with a learning effect/practice effect in your study (see below). I suggest you include a brief version of this answer, with reference to literature, in your limitations section. Just to make sure that no one will have  doubts regarding your methodology.     

Answer 1: Reviewers bring up a very important topic that we feel was not fully discussed in our paper. Belinger et al. (2005) examined practice effects in 28 healthy adults.  Neuropsychological tests were administered in a fixed order for 6 weeks. Results reveal that significant improvements were observed for all tests except the Trail B test. Our methodology is also similar to a paper published by Rogerson et al. (2016), who measured cognition (Digit test) after 15 minutes of cycling indoor or outdoor. An earlier paper from the same team (Rogerson et al., 2015) also measured cognition before and after 15 min of aerobic exercise on a treadmill using virtual reality. Another literature that supports the exclusion of learning effect/performance bias is the acute effects of interrupting prolonged periods of sitting with physical activity on cognition field. Chueh et al., (2022) have recently published a systematic review on this topic. These studies measure cognitive function numerous times throughout the day and do not show a learning effect or performance bias. Bergouignan et al. (2016) found that in 30 adults, there was no improvement of the Trail Making Test when regularly tested over a 6-hour day. Wennberg et al. (2016) have also measured cognition throughout a 7-hour day in overweight and obese individuals. They do not report any improvement on working memory task (the digit is a working memory task) in this population. Also, our data indicated that an increased frequency of the test did not yield significantly better accuracy, demonstrating that there was not a substantial learning effect associated with the way this task was administered.

Author Response

Comment 1: Nicely done. Most of my concerns has been addressed and I think the quality of the paper has improved. I have one final suggestion. In your response to my previous comments you explained in detail why you are not overly concerned with a learning effect/practice effect in your study (see below). I suggest you include a brief version of this answer, with reference to literature, in your limitations section. Just to make sure that no one will have  doubts regarding your methodology.     

 Answer 1: Thank you very much for your time and your feedback. We have incorporated a section on the learning effect in our limitation section.

Another limitation of our study is that cognitive tasks were performed relatively close to each other, thus making it impossible to rule out a possible learning effect. Numerous studies that have repeatably measured cognition (memory and executive function), throughout the day or during a short period of time, report no learning effect/performance bias (37-38). That said, our data indicates that an increased frequency of the test did not yield significantly better accuracy, demonstrating that there was not a substantial learning effect associated with the way this task was administered.

Reviewer 3 Report

Thank you for incorporating some of my comments and feedback. The discussion section has significantly improved in my view! While I still feel that the paper is trying to claim too much given its limited scope and preliminary nature (cognitive performance and how this might be a window for cognitive health is not entirely clear for me - what do you hope to improve and how might this happen via nature walks and what does it have to do with cognition?), I do see that it is covering an important topic that deserves attention. I already made this clear in the first review already, I hope.

However, cognitive performance in the sense of executive function and working memory (which aspects, why should they improve and how?) has still not been discussed appropriately. There is no explanation given as to why you chose these tests and what you were expecting. Just stating that " we look to respond to the lack of research on the potential cognitive effects of nature walks versus a nature walk in virtual reality..." (look is again not the right word here I believe) doesn't provide information as to the tests you chose, why you didn't have a control group that didn't do any walk, for instance. I think these issues need to be addressed- especially also the lack of a control group should be discussed.

Some minor issues still - please carefully read the passages that have been changed, I believe that sometimes due to deleted phrases or parts, there are incomplete sentences.

Introduction:
The first sentence of the introduction doesn't make sense - rephrase  (and maybe replace 'though' with 'although'/'despite documented positive effects' to be a little bit more formal)

p. 2/13 line 32 - incomplete sentence now

Author Response

Reviewer 2 :
Comment 2: However, cognitive performance in the sense of executive function and working memory (which aspects, why should they improve and how?) has still not been discussed appropriately.

Answer 2: The reviewer bring up a very good point. Numerous studies have shown the role of acute physical activity on cognition. Particularly, these studies have demonstrated that acute aerobic exercise can lead to more improvements in tasks that require executive functioning and memory. Other studies have shown a relationship between exercise, executive function and academic achievement test. Hillman et al. (2009) revealed that a single acute bout of aerobic exercise could improve executive function and ERP response. This also led to an increase in academic achievement test. They conclude that aerobic exercise might serve as a cost-effective mean for improving specific aspects of academic achievement and enhancing cognitive function

We have added this paragraph in our study design:

The selection of the cognitive tests is based on previous studies that reveal a positive relationship between an acute bout of aerobic exercise, executive function, and academic achievement test (22).

We have also added this paragraph in our Discussion:

Other studies have shown a relationship between exercise, executive function, and academic achievement test. Hillman et al. (2009) revealed that a single acute bout of aerobic exercise could improve executive function and neural responses. This improvement also led to an increase in academic achievement test. They conclude that aerobic exercise might serve as a cost-effective mean for improving specific aspects of academic achievement and enhancing cognitive function (22).

Comment 3: There is no explanation given as to why you chose these tests and what you were expecting.

Answer 3 : We added a section from line 366-371 which can answer this comment.

We also added a paragraph in the study design section which can help justify the use of tests.

Comment 4:Just stating that " we look to respond to the lack of research on the potential cognitive effects of nature walks versus a nature walk in virtual reality..." (look is again not the right word here I believe) doesn't provide information as to the tests you chose,

Answer 4: This has been addressed . Thank you.

Comment 5: I think these issues need to be addressed- especially also the lack of a control group should be discussed.

Answer 5 : We thank the reviewer for his very important comment concerning the control group. We have added a paragraph from line 511-516 in our limitation section which recognizes the limitation of having no control group.

Given the exploratory nature of this pilot study, no control group data was collected. Instead, we structured the study in such a way as to measure and compare levels of cognitive performance before and after two variations of a nature walk within a single group. We fully understand that choosing not to add a control group to the methodological design represents an important limitation and recommend that future studies consider a design structure that involves a control group.

Comment 6: Some minor issues still - please carefully read the passages that have been changed, I believe that sometimes due to deleted phrases or parts, there are incomplete sentences.

Answer 6: Thank you for your comment. We went through the whole document for incomplete sentences.

All minor comments have been addressed.